# Molecular and Phenotypic Changes in FLExDUX4 Mice

**DOI:** 10.3390/jpm13071040

**Published:** 2023-06-25

**Authors:** Kelly Murphy, Aiping Zhang, Adam J. Bittel, Yi-Wen Chen

**Affiliations:** 1Institute for Biomedical Sciences, The George Washington University, Washington, DC 20037, USA; 2Center for Genetic Medicine Research, Children’s National Hospital, Washington, DC 20010, USA; 3Department of Genomics and Precision Medicine, School of Medicine and Health Science, The George Washington University, Washington, DC 20037, USA

**Keywords:** facioscapulohumeral, muscular, dystrophy, FSHD, *FLExDUX4*, DUX4, muscle, TDP-43

## Abstract

Facioscapulohumeral muscular dystrophy (FSHD) is caused by the aberrant expression of the double homeobox 4 (*DUX4*) gene. The *FLExDUX4* mouse model carries an inverted human *DUX4* transgene which has leaky *DUX4* transgene expression at a very low level. No overt muscle pathology was reported before 16 weeks. The purpose of this study is to track and characterize the *FLExDUX4* phenotypes for a longer period, up to one year old. In addition, transcriptomic changes in the muscles of 2-month-old mice were investigated using RNA-seq. The results showed that male *FLExDUX4* mice developed more severe phenotypes and at a younger age in comparison to the female mice. These include lower body and muscle weight, and muscle weakness measured by grip strength measurements. Muscle pathological changes were observed at older ages, including fibrosis, decreased size of type IIa and IIx myofibers, and the development of aggregates containing TDP-43 in type IIb myofibers. Muscle transcriptomic data identified early molecular changes in biological pathways regulating circadian rhythm and adipogenesis. The study suggests a slow progressive change in molecular and muscle phenotypes in response to the low level of DUX4 expression in the *FLExDUX4* mice.

## 1. Introduction

Facioscapulohumeral muscular dystrophy (FSHD) is an autosomal dominant disorder caused by the aberrant expression of the double homeobox 4 (*DUX4*) gene, resulting in progressive muscle loss. Symptoms of FSHD include weakness in the muscles of the face, shoulders, and upper arms, and can include limb and trunk muscles as the disease progresses [1,2,3]. It has a prevalence of 1 in 8333–20,000 people in the world [1,4,5,6,7]. Current therapies for FSHD only address its symptoms, such as shoulder fixation orthotics and physical therapy [2].

FSHD is subclassified into two subtypes, FSHD type 1 and FSHD type 2, which affect approximately 95% and 5% of individuals with FSHD, respectively [1,2]. FSHD type 1 and type 2 are clinically indistinguishable but are classified by different genomic mutations that affect the regulation of epigenetic control over the D4Z4 region. FSHD1 is linked to a contraction of the tandem D4Z4 repeat array on chromosome 4q35 to 1–10 repeats [8,9]. In unaffected individuals, this array contains 11–150 copies of the D4Z4 repeat. FSHD2 is reported to be caused by mutations in the structural maintenance of chromosomes flexible hinge domain containing 1 (*SMCHD1*) gene located on chromosome 18, in the DNA-methyltransferase 3 beta gene (*DNMT3B*) located on chromosome 20, or in ligand-dependent nuclear receptor-interacting factor 1 (*LRIF1*) on chromosome 1 [6,10,11]. Contraction of the D4Z4 array, haploinsufficiency of SMCHD1, mutations in *DNMT3B*, or mutations in *LRIF1* are reported to be associated with decreased DNA methylation and transcriptionally de-repress the *DUX4* gene [6,10,12,13,14,15,16,17,18,19,20]. These genomic mutations cause an indistinguishable clinical presentation because they all epigenetically de-repress the *DUX4* gene located in the D4Z4 repeats. Previous studies reported that the development of FSHD requires a combination of two genomic features to develop FSHD. In addition to the genomic mutations mentioned above, the second genomic feature necessary for FSHD pathogenesis is the presence of a functional polyadenylation signal in the pLAM region distal to the D4Z4 array [21,22,23,24,25]. This polyadenylation signal allows for the stabilization of the *DUX4* mRNA transcribed from the last D4Z4 repeat for the translation of the DUX4 protein [21,22,23,24,25]. The functional polyadenylation signal, in addition to the genomic mutations listed above, allows for the transcription and translation of the *DUX4* mRNA and protein, which causes FSHD.

Previous studies have shown that the DUX4 protein is a transcription factor, which is expressed during early embryonic development or postnatally in the testes, thymus, mesenchymal stem cells, and keratinocytes [26,27,28,29,30,31,32]. The proposed function of DUX4 is that it facilitates early genome activation, also known as the zygotic genome activation, at the two- to four-cell stage of development [26,27,28,33]. In FSHD, the aberrant expression of DUX4 in the muscle causes the expression of germline DUX4 transcriptional targets [26]; the repression of oxidative stress response genes [34,35,36,37]; disruptions in cell cycle and migration [35,38]; disruption of RNA metabolism, splicing, surveillance, and transport [38,39]; and a reduction in myogenic capacity due to the misregulation of MYOD1 and its downstream targets, including structural and contractile components [28,29,33,36,37,40,41,42,43]. The culmination of the effect of DUX4 expression on these pathways results in progressive muscle weakness and loss in individuals with FSHD.

Historically, mouse models for FSHD have had limited experimental use due to either severe phenotypes preventing long-term studies or a lack of detectable *DUX4* expression and muscle pathologies [25,44,45,46,47]. The *FLExDUX4* model carries an inverted human *DUX4* transgene flanked by *lox* sites, which, when crossed with the *ACTA1-MCM,* allows for tunable *DUX4* expression in the skeletal muscle of mice [48,49,50]. Induction of the *DUX4* transgene in the *ACTA1-MCM/FLExDUX4* mice led to FSHD-like phenotypes, but required repeated injections of tamoxifen to maintain moderate to severe levels of pathology [48,49]. The single transgenic *FLExDUX4* mouse model, while presenting with no overt pathology at a young age, showed leaky expression of *DUX4* despite the inverted transgene [48,49]. The absence of pathology compared to the conditional counterparts have left the *FLExDUX4* mice relatively under studied. Since patients with FSHD have a wide range of clinical presentations and age of onset, it is important to thoroughly characterize all iterations of the *FLExDUX4* model for their pre-clinical relevancy [19,51,52]. In this study, we hypothesized that the long-term expression of a low level of *DUX4* will lead to muscle pathologies and functional deficits in older *FLExDUX4* mice. To explore this, we investigated phenotypic (e.g., body and muscle weight), functional (e.g., grip strength), pathological (e.g., fiber size and typing), and molecular changes (e.g., transcriptome) in the *FLExDUX4* mice at various ages to determine changes during disease progression.

## 2. Materials and Methods

### 2.1. IACUC Statement

Animal experiments were approved by the Institutional Animal Care and Use Committee at Children’s National Hospital in Washington, DC. Euthanasia was performed using carbon dioxide asphyxiation, and death was ensured with cervical dislocation. For all experiments, hemizygous B6(Cg)-Gt(ROSA)26Sortm1.1(DUX4*)Plj/J (FLExDUX4) mice were used.

### 2.2. Genotyping

Genomic DNA was isolated from tail snips using phenol–chloroform extraction. Tail snips were digested in 300 µL digestion buffer (50 mM Tris-HCl pH 7.4, 5 mM EDTA pH 8.0, 0.5% sodium dodecyl sulfate) containing 10 µL proteinase K (20 mg/mL; Ambion, Huntingdon, UK). Tails were incubated overnight at 55 °C with shaking at 300 rpm. Debris was pelleted by centrifugation at 21,000× *g* for 30 s and the supernatant collected. Phenol–chloroform–isoamyl alcohol (25:24:1) was added to the supernatant in a 1:1 ratio. Samples were inverted by hand for 15 s and centrifuged at 21,000× *g* for 2 min. The aqueous phase was collected and added to ammonium acetate (150 µL at 7.5 M) and 100% ethanol (750 µL, −20 °C). Samples were centrifuged at 12,000× *g* at room temperature for 20 min. The pellet was washed twice with 500 µL of chilled 80% ethanol and centrifuged at room temperature at 21,000× *g* for 5 min. Pellets were air dried for 15 min and re-suspended in 50 µL of water. Sample concentration was determined with a NanoDrop spectrometer and diluted to 100 ng/µL.

PCR amplification of the *DUX4* transgene was performed according to published study [48,49], using 20 ng of DNA, 2.74 µL GoTaq Hot Start Polymerase master mix (Promega, Madison, WI, USA), and 400 nM of the *Rosa26/FLExDUX4* primers pair or 200 nM of the wild type *Rosa26* primers. The reaction volume was brought to 12 µL with nuclease-free water. Primer sequences were *Rosa26* (forward): 5′-CAATACCTTTCTGGGAGTTCTCTGCTGC-3′; *Rosa26* (reverse): 5′-TGCAGGACAACGCCCACACACC-3′; and *Rosa26/FLExDUX4* (reverse): 5′-CTCGTGTAGACAGAGCCTAGACAATTTGTTG-3′. Reactions were performed with a pre-PCR hold at 94 °C for 3 min, then cycled 35 times (94 °C for 20 s, 62 °C for 20 s, and 72 °C for 35 s) and a final extension at 72 °C for 2 min. Gel electrophoresis was performed using a 2% agarose gel using the ZipRuler express DNA ladder to determine band size (ThermoFisher Scientific, Waltham, MA, USA). Wild-type mice had only the 175 bp product produced by the *Rosa26* primers. Hemizygous mice had both the 409 bp amplicon of the *Rosa26/FLExDUX4* pair and the 175 bp amplicon of the *Rosa26* primers [48,49].

### 2.3. RNA Isolation and RT-qPCR

For muscle, total RNA was isolated from the quadriceps and triceps. Muscles were homogenized in 1 mL of TRIzol, and debris was pelleted out by centrifuging samples at 11,600× *g* for 5 min. Two hundred microliters of chloroform was added to the supernatant, and the samples were vortexed for 15 s. Samples were incubated at room temperature for 3 min and subsequently centrifuged at 4 °C and 11,600× *g* for 15 min. The aqueous phase was collected and added to 0.5 mL of isopropyl alcohol. Samples precipitated at room temperature for 10 min and centrifuged at 4 °C and 11,600× *g* for 10 min. Pellets were washed twice with 1 mL of chilled 75% ethanol and centrifuged at 7500× *g* for 5 min. Pellets were air dried for 15 min. RNA was purified with the RNeasy Micro kit with DNase digestion (Qiagen, Hilden, Germany).

Complementary DNA (cDNA) was synthesized using the SuperScript IV Reverse Transcriptase kit (ThermoFisher Scientific). Samples were prepared in a 13 µL reaction consisting of 2 µg of total RNA, 1 µL dNTPs (10 mM stock; New England Biolab, Ipswich, MA, USA), 1 µL oligo(dT)12–18 (Life Technologies, Carlsbad, CA, USA), and nuclease-free water and incubated at 65 °C for 5 min. Samples were then chilled on wet ice for 7 min. A solution of 4 µL 5× first strand buffer, 1 µL DTT, 1 µL of SuperScript IV (ThermoFisher Scientific), and 1 µL RNasin (Promega) was added to each reaction and incubated for 25 °C for 5 min, 50 °C for 1 h, and 70 °C for 15 min. Sample cDNA was brought to a volume of 100 µL.

Quantitative PCR was performed in triplicate using the SYBR Green Master Mix (Applied Biosystems, Waltham, MA, USA), 1 µL of cDNA, and 200 µM of forward and reverse primers. The total reaction volume was 20 µL. Reactions were performed with one pre-PCR hold at 50 °C for 2 min, one pre-PCR hold at 95 °C, and 40 cycles of amplification (95 °C for 15 s and 60 °C for 1 min). Relative gene expression was analyzed with the delta delta ct method. The following genes were assayed: *DUX4,* F-box protein 32 (*Fbxo32*), and muscle RING-finger protein-1 (*Murf1*). Primers Glyceraldehyde-3-phosphate dehydrogenase (*Gapdh*) and Hypoxanthine-guanine phosphoribosyltransferase (*Hprt1)* served as internal controls for *Fbxo32,* and *Murf1*. Primers used were *Fbxo32/Atrogin1* (forward): 5′-TCAGAGAGGCAGATTCGCAAGC-3′ and (reverse): 5′-GTCAGTGCCCTTCCAGGAGA-3′; *Murf1* (forward): 5′-TTGACTTTGGGACAGATGAGG-3′ and (reverse): 5′-AGCGTGTCTCACTCATCTCCTT-3′. Primers for internal controls were *Gapdh* (forward): 5′-TTGTCAGCAATGCATCCTGC-3′ and (reverse): 5′-CCGTTCAGCTCTGGGATGAC-3′ [53]; *Hprt1* (forward): 5′-CGTCGTGATTAGCGATGATG-3′ and (reverse): 5′-TTTTCCAAATCCTCGGCATA-3′ [41].

### 2.4. Muscle Collection

The gastrocnemius, soleus, tibialis anterior, quadriceps, deltoid, triceps, bicep, masseter, diaphragm, and heart muscles were removed and weighed. Muscles were snap-frozen in isopentane cooled in liquid nitrogen for RNA and protein processing.

### 2.5. Grip Strength Measurement

Grip strength measurements were performed using a bar and a grid connected to an isometric force transducer (Columbus Instruments, Columbus, OH, USA). Briefly, *FLExDUX4* mice were acclimated with five pulls five times each for both forelimb and hindlimb over three days. Grip strength measurements occurred over five days and each mouse performed five pulls for each measurement [54]. Measurements were taken on a predetermined location on the force meter to prevent measurement differences as a result of location on the attached grid. The strongest pull from each day was collected and averaged [54]. Student’s *t*-test was used to compare the experimental groups to the control groups. Grip strength testing was performed at 5 months, 8 months, and 12 months of age.

### 2.6. Hematoxylin and Eosin Staining

Snap-frozen muscles were sectioned at 8 µm using a Leica CM1950 cryostat and air-dried overnight at room temperature. Sections were stained with Modified Mayer’s hematoxylin for 30 min. Slides were dipped twice in distilled water and dipped in acid alcohol five times. Slides were rinsed in distilled water for 1 min and stained with eosin for 3 min. Slides were washed in two exchanges of 95% ethanol for 2 min with a final wash in 100% ethanol for 2 min. Slides were then washed in two exchanges of xylene for 2 min and 5 min. Coverslips were mounted with Permount mounting medium (Fisher Scientific). The sections were imaged with a VS120 Olympus scanning microscope.

### 2.7. Nicotinamide Adenine Dinucleotide Tetrazolium Staining

Frozen quadriceps sections were prepared for NADH-TR staining as described previously [54]. Briefly, 8 µm sections were prepared then incubated in a Tris Buffer (0.05 M, pH 7.6 (Millipore Sigma, Burlington, MA, USA) solution containing nitro-blue tetrazolium (2 mg/mL, Sigma) and NADH (8 mg/5 mL, Sigma) at 37 °C for 30 min. Sections were washed 3 times with deionized water, followed by 3 exchanges each of 30%, 60%, 90%, 60%, and 30% of acetone (1 min each). Slides were rinsed 3 times with deionized water and mounted with aqueous media [54].

### 2.8. BODIPY Staining

Cryosections (5 µm) were dried at room temperature for 2 h. BODIPY™ 493/503 was used to stain neutral lipids (0.25 µg/mL, ThermoFisher Scientific, Cat#D3922). Hoechst 33342 Solution was used to stain nuclei (0.1 µg/mL). Sections were incubated in solution made of 1× PBS with BODIPY and Hochest for 15 min. Sections were washed five times for 3 min in 1× PBS. The whole sections were imaged with a VS120 Olympus microscope. Images were analyzed with Image J.

### 2.9. Immunohistochemistry

Frozen quadriceps muscles were sectioned at 5 µm and stained using the Vectastain ABC HRP kit (Peroxidase) for mouse or rabbit antibodies (Vector Laboratories, Newark, CA, USA). Frozen muscle sections were fixed in cold acetone for 10 min and air-dried for 5 min. Sections were then blocked in a solution of 0.3% H_2_O_2_ and 0.3 Normal sera in 1× PBS for 5 min. Slides were washed twice with 1× PBS for 5 min. For TDP-43 staining, slides were blocked for 30 min according to the manufacturer’s protocol. Sections were incubated with the TDP-43 primary antibody (1:200, rabbit, Proteintech, Rosemont, IL, USA) overnight at 4 °C. For myosin heavy chain staining, slides were blocked in Vectastain blocking reagent overnight, then incubated with primary antibodies myosin heavy chain type 1 (A4.951), type 2a (2F7), type 2B (10F5), or type 2 × (6H1) (2 µg/mL, mouse, Developmental Studies Hybridoma Bank, The University of Iowa, Iowa City, IA, USA) for 2 h at room temperature [55,56]. DSHB Hybridoma Product A4.951 was deposited by Helen Blau of Stanford University [55,56]. DSHB Hybridoma products 2F7, 10F5, and 6H1 were developed and deposited by Christine Lucas of the University of Sydney [55,56].

After incubation with the primary antibody, all sections were washed 3 times for 15 min in 1× PBS. Sections were incubated with secondary antibodies for 1 h and rinsed 3 times for 15 min in 1× PBS. Slides were developed in DAB reagent for 10 min and washed with deionized water for 5 min. Slides were then washed twice in 1× PBS for 10 min. Slides were mounted with Crystal Mount (Electron Microscopy Sciences).

### 2.10. Immunofluorescence Staining

Frozen quadriceps were sectioned at a 5 µm thickness and dried at room temperature for 1 h. Slides were then blocked using filtered blocking buffer containing 10% horse serum (Gibco), 1% normal goat serum, and mouse on mouse (Vector Labs) in 1× PBS for 1 h. Primary antibodies were added in antibody buffer (10% horse serum and 1% goat serum in 1× PBS) for 2 h at room temperature. Slides were washed three times for 15 min in 1× PBS. Secondary antibodies were diluted in antibody buffer and incubated on the slides for 1 h at room temperature. Slides were washed in 1× PBS three times for 15 min. Primary antibodies used were Dystrophin (1:100, Abcam, Cambridge, UK) and myosin heavy chain antibodies type 1 (BA-D5), type 2a (SC-71), type 2b (BF-F3), and type 2 × (6H1) (2 µg/mL, Developmental Studies Hybridoma Bank) [57,58]. Clones BA-D5, SC-71, and BF-F3 were deposited to the DSHB by Schiaffino, S. (DSHB Hybridoma Products BA-D5, SC-71, and BF-F3) [57,58].

Secondary antibodies used were DyLight™ 405 AffiniPure Goat Anti-Mouse IgG (1:500, Jackson Immunoresearch Laboratories, Inc., West Grove, PA, USA), Cy™3 AffiniPure Goat Anti-Mouse IgM, µ chain specific (1:500, Jackson Immunoresearch Laboratories, Inc.), Alexa Fluor 647 AffiniPure Goat Anti-Rabbit IgG (H + L) (1:500, Jackson Immunoresearch Laboratories, Inc.), and Alexa Fluor 488 goat anti mouse IgG (1:1000, ThermoFisher Scientific). Slides were washed 3 times in 1× PBS for 15 min and mounted with Fluoromount mounting media (Electron Microscopy Sciences).

### 2.11. Fiber Size Data Analysis

Quadriceps sections were imaged using a PURNA-Olympus VS-120 scanning microscope, and images were analyzed using Image J (NIH, https://imagej.net/ij/index.html (accessed on 13 June 2023)). Positive fibers were identified after normalizing all sections to the same threshold. The threshold was determined at the point at which fibers with the light blue background stain were eliminated. The minimal Feret’s diameter was determined using the middle section of the quadriceps located between two connective tissue landmarks present in all images (586–799 fibers in each section). For fluorescent staining, images were processed using the ImageJ software, and the minimal Feret’s diameter and fiber area were determined using fiber boundary visualized by dystrophin staining. The fibers measured where located within the same landmarks as the NADH-TR sections. The following numbers of fibers were measured: 256–386 type 2a fibers, 200 type 2b fibers, and 383–539 type 2x fibers in each section. ImageJ calculated the circularity of each fiber using the following formula: circularity = 4pi (area/perimeter^2^). A value of 1.0 denotes a perfect circle and an elongated polygon as the value approaches 0.0 (https://imagej.nih.gov/ij/plugins/circularity.html (accessed on 13 June 2023)). Five random fields were taken of the whole section. The minimal Feret’s diameter was determined by measuring the fiber boundary visualized by dystrophin staining. A range of 3538–4665 fibers was measured in each section.

All fibers were ranked based upon their size, with the smallest fiber assigned the number one. The rank number was normalized by dividing the rank by the total number of fibers to obtain a value between 0 and 1. Normalized rank was plotted against the fiber diameter or fiber area. The Kolmogorov–Smirnov test was used to determine the significant difference in the fiber size distribution between littermate pairs as previously described [54,59].

### 2.12. RNA-Sequencing

Total RNA from 2-month-old male *FLExDUX4* mice was isolated from triceps muscle using the TRIzol/Qiagen protocol described above (*n* = 3). Samples were processed by Lexogen GmbH for transcriptome sequencing (Vienna, Austria). Sample RNA was checked for quality using the Nanodrop2000c (ThermoFisher Scientific) and integrity with the Fragment Analyzer high-sensitivity assay (Agilent). The library was sequenced using the QuantSeq 3′mRNA-seq Library Prep kit FWD for Illumina (015UG009V0252) using the standard QuantSeq-FWD protocol. Sequencing was performed on an Illumina NextSeq 500 sequencer with a v2 SR75 High Output kit. Cutadapt version 1.16 was used to remove adapter contaminations, continuous polyA sequences, and continuous polyG sequences at the 3′ end. Reads were aligned to the mouse Mmu_GRCm38.90_ERCC_SIRV genome reference (Genome Reference Consortium) using the STAR tool (spliced transcripts alignment to a reference) version 2.5.3a. Normalization and differential expression analysis was performed with DESeq2 version v1.18.1 with significance determined at *adj p* < 0.1. Top pathways were determined using Ingenuity Pathway Analysis (IPA).

## 3. Results

### 3.1. FLExDUX4 Mice Show Muscle Weakness Measured by Grip Strength Testing

To determine if the *FLExDUX4* mice showed muscle weakness, the grip strengths of a cohort of *FLExDUX4* mice and their wild-type litter mates (male *n* = 8–9, female *n* = 3–4) were measured using an isometric force transducer at 5, 8, and 12 months of age (Figure 1). At all-time points there was a significant difference in the force measurement between the *FLExDUX4* and wild-type male (Figure 1A,C) and female mice (Figure 1B,D) (*p* < 0.05). At 5 months old, male *FLExDUX4* mice were an average of 25% weaker in the forelimb (*p* < 0.001) and 17% weaker in the hindlimb (*p* < 0.01). At 8 months old, male *FLExDUX4* mice were an average of 22% weaker in the forelimb (*p* < 0.001) and 13% weaker in the hindlimb (*p* < 0.05). At 12 months old, male *FLExDUX4* mice were 21% weaker in the forelimb (*p* < 0.001) and 17% weaker in the hindlimb (*p* < 0.05). In female *FLExDUX4* mice, the forelimb strength was an average of 18% weaker (*p* < 0.05), and the hindlimb was an average of 20% weaker at 5 months old (*p* < 0.05). At 8 months old, female *FLExDUX4* mice were an average of 15% weaker in the forelimb (*p* < 0.05) and 23% weaker in the hindlimb (*p* < 0.01). At 12 months old, female *FLExDUX4* mice were an average of 22% weaker in the forelimb (*p* < 0.001) and 30% weaker in the hindlimb (*p* < 0.05).

### 3.2. FLExDUX4 Mice Display Sex Differences in Body Weight and Muscle Weight

Body weight differences in mice approximately 20 weeks old were previously reported in *FLExDUX4* mice [48]; however, muscle weight data were not reported. In this study, we measured the body weight and muscle weight in several cohorts of mice at 2 months, 4 months, 8 months, and 12 months to determine weight changes in the *FLExDUX4* mice when they age. The muscles examined include the gastrocnemius, soleus, tibialis anterior, quadriceps, deltoid, triceps, biceps, masseter, diaphragm, and heart. Both male (*n* = 6–9) and female (*n* = 6–8) *FLExDUX4* mice showed significant differences in body weight (Figure 2) and muscle weight (Appendix A) compared to their wild-type littermates. However, the body weight difference appeared at a much younger age in the male mice (4 months old) compared to the female mice (12 months old). Male *FLExDUX4* mice showed body weight differences at 4 months of age and were an average of 3.8 g lighter (11%, *p* < 0.05). At 8 months old, the *FLExDUX4* mice were an average of 9.7 g (24%, *p* < 0.05) lighter. At 12 months old, male *FLExDUX4* mice were an average of 14.7 g (31%, *p* < 0.05) lighter than their wild-type littermates (Figure 2A). Unlike their male counterparts, female *FLExDUX4* mice did not show a significant difference in body weight until 12 months old. At 12 months old, female *FLExDUX4* mice were an average of 9.3 g lighter (26%, *p* < 0.05) than their wild-type littermates (Figure 2B).

The muscle weight data showed that the muscle weight of male *FLExDUX4* mice was significantly lower compared to their wild-type littermates at 2 months, 4 months, 8 months, and 12 months old (Appendix A). In male *FLExDUX4* mice, significant differences in the weights of three hindlimb muscles (gastrocnemius, soleus, and quadriceps) were identified at 2 months, 4 months, and 12 months old (*p* < 0.05). The tibialis anterior muscles showed significant differences at 2 months and 4 months old (*p* < 0.05). Significant differences in the biceps and triceps of the forearm were observed at 12 months old (*p* < 0.05). The muscles most frequently affected were the hindlimb muscles: gastrocnemius, soleus, tibialis anterior, and the quadriceps. The deltoid, masseter, heart, and diaphragm were the most frequently spared. In female *FLExDUX4* mice, many of the muscles did not show significant differences until 12 months of age (Appendix A). The muscles that showed significance were the deltoid, masseter, triceps, and quadriceps at 4 months old (*p* < 0.05). The masseter was the only muscle affected at 8 months of age. At 12 months old, the deltoid, triceps, biceps, gastrocnemius, soleus, tibialis anterior, and the quadriceps were significantly smaller in *FLExDUX4* mice (*p* < 0.05). Additionally, muscle weight in both male and female *FLExDUX4* mice and their wild-type littermates increased over time. At 12 months of age, wild-type littermates had higher or similar muscle weights compared to the 8-month-old wild-type mice. However, the *FLExDUX4* mice showed a decrease in muscle weight.

### 3.3. Fiber Size Variations Were Observed in Type IIa and IIx Muscle Fibers in Older FLExDUX4 Mice

Because the male mice showed significant muscle weight differences over time, we further examined muscle pathologies in the male *FLExDUX4* mice to determine what contributed to the muscle weight loss. To examine histological changes in *FLExDUX4* mice at young and old ages, quadriceps from 2-month-old and 12-month-old male mice were studied. At 2 months old, there was no overt pathology found in the *FLExDUX4* mice compared to their wild-type littermates. No significant difference was observed in myofiber size in 2-month-old *FLExDUX4* mice. In 12-month-old mice, a deviation in myofiber size (Feret’s diameter) was observed in the smallest 40% of fibers measured, but no difference was observed in the largest fibers measured (Appendix A).

The presence of small myofibers is often reported in diseased muscle. This can be due to regeneration of muscle fibers or muscle fiber atrophy [54]. Analysis of the number of central nucleated fibers did not show a significant difference between the *FLExDUX4* and wild-type quadriceps. NADH-TR staining has been used to characterize myofiber types and visualize angular atrophic myofibers [54,60]. Atrophic myofibers are angular-shaped small fibers heavily stained with NADH-TR. To determine if atrophic fibers were present, the 12-month-old quadriceps were stained with NADH-TR. The *FLExDUX4* mice presented with smaller NADH-TR positive muscle fibers, and some of them were angular-shaped (Figure 3 and Appendix A). Two genes, *Atrogin 1* and *Murf1*, that are commonly involved in muscle atrophy were observed by RT-qPCR, and no significant differences in expression were detected.

Muscle fibers can be subdivided into the oxidative (type I) and glycolytic (type II) myofibers. In mice, a gradient exists between the glycolytic fibers, where the larger type IIb fibers are more glycolytic (light blue with NADH-TR) and smaller type IIa fibers are more oxidative (dark blue with NADH-TR) [60,61,62]. The deviation in fiber sizes suggested that different fiber types were affected in 12-month-old *FLExDUX4* mice (Figure 3 and Appendix A).

To determine how specific fiber types are affected in the *FLExDUX4* mice, quadriceps from 2-month-old and 12-month-old *FLExDUX4* mice were co-stained with myosin heavy chain type I, type IIa, and type IIx. Each myofiber was measured to determine fiber size and circularity. Circularity measurements were used in this experiment as a means to quantify myofiber shape. Atrophic myofibers have been shown to be more angular and therefore would have a lower circularity value compared to a healthy myofiber. The data showed that 2-month-old mice did not show significant differences in size or circularity of each muscle fiber type. At 12 months old, the *FLExDUX4* mice had significantly smaller type IIa (*p* < 0.05) and type IIx (*p* < 0.05) fibers (Figure 4). There was no significant difference in the size of the type IIb fibers. Fiber circularity of the type IIx fibers showed significant differences, indicating that the fibers are more angular-shaped (Figure 5). These data showed that, over time, the oxidative subset of the type II myosin heavy chain fiber types became smaller and atrophic.

### 3.4. TDP-43-Positive Aggregates Were Found in the Type IIb Fibers

In addition to the presence of smaller fibers in the 12-month-old *FLExDUX4* mice, hematoxylin and eosin-stained sections of the quadriceps muscles appeared to have basophilic aggregates (Figure 6). These aggregates were consistent in appearance to a mouse model that overexpresses TAR DNA-binding protein 43 (TDP-43), and the aggregation of TDP-43 has been observed in FSHD myoblasts in vitro in response to *DUX4* expression [63,64,65]. Based on data from the immunochemistry staining of serial sections, the aggregates stained were TDP-43-positive and located specifically in the type IIb fibers.

### 3.5. RNA Sequencing Revealed Changes in Metabolism and Oxidative Stress Response

To determine molecular changes before the muscle pathology became apparent, we conducted RNA-seq of muscles from 2-month-old male mice. At 2 months of age, there were 573 differentially expressed genes (*p* < 0.05) in the comparison between the *FLExDUX4* vs. wild-type mice.

We used ingenuity pathway analysis (IPA) to determine which genes are enriched in specific functional groups of genes based on the sequencing data. The IPA revealed changes in several pathways involved in growth and muscle remodeling. At two months of age, the top three ranked pathways were Adipogenesis (*p* < 0.001), circadian rhythm (*p* < 0.001), and NRF2-mediated oxidative stress response (*p* < 0.001) (Appendix A). Canonical circadian rhythm genes, namely *Clock*, *Cry2*, and *Per2,* were shown to be differentially expressed in *FLExDUX4* mice [66]. The NRF2 pathway has also been reported to be Clock-controlled and has been shown to have differential expression in FSHD [35,67,68]. In the adipogenesis pathways, Forkhead box protein O1 (*Foxo1*) and Lipoprotein lipase (*Lpl*) were both upregulated. The *Foxo1* gene was reported to inhibit adipocyte differentiation and initiate muscle atrophy. The activation of *Lpl* by Foxo1 facilitates a metabolic switch from carbohydrate oxidation to lipid oxidation in muscle [35,67,68,69,70,71,72]. Indeed, BODIPY staining of quadriceps muscle from 4-month-old *FLExDUX4* mice showed significant increases in intermuscular fat in the muscle (Appendix A). Three metabolic pathways identified based on the IPA (Methylmalonyl, 2-oxobutanoate degradation I, PPARα/RXRα activation) play a role in lipid metabolism and generation of components for the Krebs cycle [73]. The PPARα/RXRα activation in addition to CD27 signaling was reported to play a role in inflammatory pathways [73]. Additional pathways identified include genes in changes in pathways involved in muscle growth (AMPK signaling), and pathways elevated in oxidative stress response (NRF-2 mediated response). These molecular changes were identified in muscles without overt pathologies.

## 4. Discussion

Most mouse models for FSHD have been limited in experimental use due to either severe phenotypes preventing long-term studies or inconsistent *DUX4* expression and pathologies [25,44,45,46,47]. Previously published data on the *FLExDUX4* single transgenic animals showed that the *FLExDUX4* mice showed body weight differences, had leaky expression of *DUX4* mRNA, and did not show overt muscle pathology in 23-week-old female *FLExDUX4* mice [48]. A more recent study reported that the *FLExDUX4* mice have a mild but significant increase in fibrosis at 6 months old [74]. The goal of this project was to track changes in muscle weight, functional, and muscle pathologies in response to the leaky low-level expression of *DUX4* for a long period of time. We found that the *FLExDUX4* mice presented with several phenotypes as they aged. Both male and female *FLExDUX4* mice had significantly weaker grip strength measurements. Male mice presented with significantly smaller body weight and muscle weight compared to their wild type littermates. In addition, the males developed the phenotypes at a younger age compared to the female *FLExDUX4* mice. The RNA-seq data showed that *DUX4* expression in myofibers affected gene expression associated with pathways of adipogenesis, circadian rhythm, and NRF2-mediated oxidative stress response. Pathological data showed that prolonged expression of *DUX4* leads to TDP-43 positive aggregates in type IIb fibers and a decrease in the size of type IIa and type IIx fibers.

The *FLExDUX4* mice showed significant differences in body weight. Male *FLExDUX4* mice became significantly smaller once mice reached adulthood at 4 months old which is in concordance with previously reported data by Jones et al. in 20-week-old mice [48]. Different from the previously reported data, we did not observe changes in body weight of *FLExDUX4* female mice until 12 months of age [48]. Body weight and muscle weight data suggested that the *FLExDUX4* mice showed sex differences. Male *FLExDUX4* mice have significantly lower absolute muscle weight starting at 2 months of age and up to 12 months of age with a progressive increase in the number of muscles affected. Few muscles of the female *FLExDUX4* mice showed significant weight differences until 12 months old. A previous study suggested that estrogen has a protective effect in DUX4-expressing muscles [75]. Estrogen protection in the *FLExDUX4* mice may partially explain the sex differences observed in our experiment. Reproductive senescence in mice occurs between 9 and 12 months of age, which supports the age of onset of the body weight and muscle weight differences in the female *FLExDUX4* mice [76]. The data generated from the *ACTA1-cre/FLExDUX4* female mice injected with tamoxifen, an estrogen receptor inhibitor used to induce DUX4 expression, also suggested the protective effects of estrogen in the female mice. Female mice were more severely affected compared to their male littermates after *DUX4* was induced by tamoxifen injections [49].

The observation of sex differences in the body weight and muscle weight, where males are more severely affected, is similar to the observation in FSHD [41,52,77,78]. The average age of onset is by the second decade of life in males and the third decade of life in females [19,52]. In the *FLExDUX4* mice, heart, diaphragm, masseter, and deltoids were mostly spared in the aged mice, which reflects the human FSHD [77,79,80,81]. However, there was no obvious preference of forelimb vs. hindlimb muscles observed in the *FLExDUX4* mice. Various mouse models have been generated for the pre-clinical development of therapeutics for FSHD. Another inducible model, *iDUX4pA* mice, contains an inducible *DUX4* gene inserted upstream of the *HPRT* gene and under the control of the doxycycline-inducible *sgTRE* promoter [45]. The *iDUX4pA* model also displays sex differences, but male mice have only been reported to live until 4 months of age, which can limit trial length of a potential therapeutic [45]. Like the *FLExDUX4* mice, the *iDUX4pA* mice have alopecia, smaller body weights, and smaller muscle weights [45,48]. Flow cytometry and transcriptional analysis showed that the *iDUX4pA* mouse model has increased fibrosis markers in the absence of pathology, as previously observed in the *FLExDUX4* model [45,74,82]. Both models have reported leaky *DUX4* expression in other tissues [45,48]. Hearing loss and vasculature defects were observed in the *iDUX4pA* mice [45,82]. Interestingly, decreased body fat was observed in the *iDUX4pA* model as measured by MRI [45]. We did observe less body fat in the *FLExDUX4* mice; however, no quantitative data were collected. The conditional *ACTA-MCM/FLExDUX4* mice were also reported to have sex differences where the male mice were more severely affected. Conversely, when the *ACTA-MCM/FLExDUX4* mice were administered tamoxifen to induce higher DUX4 expression, the female mice are more severely affected [49]. Chronic induction and expression of *DUX4* in female *iDUX4pA* mice was reported to cause severe phenotypes, including a prominent outward spinal curvature, pronounced muscle atrophy, and muscle weakness at the end of the 6-month study [82]. Pathological changes in muscles of the mice suggest more severe phenotypes, including inflammatory infiltration, fat deposition, and deposition of extracellular matrix [82]. With the transgene induced, the *iDUX4pA* model is more comparable to the *ACTA-MCM/FLExDUX4* mice when the *DUX4* is induced by tamoxifen.

Analysis of the grip strength data showed that *FLExDUX4* mice were significantly weaker than their wild-type littermates. Previous studies reported no significant difference in time to fatigue or wire hanging assessments [48,49]. These differences may be due to age differences or differences in assays used for accessing muscle strength. The muscles of *FLExDUX4* mice do not have overt degeneration/regeneration during the observation period; however, we have previously reported an increase in endomysial fibrosis in the model [74]. Additionally, fibrosis pathways were activated in our current RNA-seq data through IPA at 2 months of age (*p* < 0.05, ranked 15), suggesting early involvement of the fibrosis pathways. Small increases in fibrosis have been shown to decrease grip strength and increase muscle weakness [83,84,85]. Muscle weakness can also occur due to additional molecular changes in the muscle [54,86,87]. Muscle biopsies from individuals with FSHD were reported to have increased expression of proteins involved in glycolysis and tricarboxylic acid cycle, lipid peroxidation, decreased mitochondrial function, and a decreased response to oxidative stress [41,88]. In our study, we showed that from a young age, genes involved in lipid oxidation and NRF-2 stress response pathways were misregulated. Exposure to reactive oxygen species has been shown to cause decreased contractility in muscle fibers [89]. The combination of fatty acid oxidation and an impaired oxidative stress response may potentially perpetuate oxidative stress that can impact functional output [35,67,68,90,91].

In this study, we identified long-term exposure to low levels of *DUX4* expression resulted in fiber-type-specific pathologies as the *FLExDUX4* mice age. Basophilic TDP-43 aggregates were observed in type IIb fibers specifically, while the type IIa and IIx fibers decreased in fiber size. The decreased fiber size of type IIa and type IIx muscle fibers in 12-month-old *FLExDUX4* muscle suggests that chronic low-level expression of DUX4 negatively affects the fiber maintenance of type IIa and type IIx. Misregulation in circadian rhythm signaling in young *FLExDUX4* mice could contribute to impaired muscle maintenance, as *MyoD1* is a documented Clock transcriptional target [92]. MyoD-regulated pathways have been reported to be affected in biopsies from individuals with FSHD [37]. Results from the NADH-TR staining showed angular-shaped myofibers with dark blue staining, which suggested these were atrophic fibers [54]; however, RT-qPCR of *Atrogin 1* and *Murf1* did not show significant differences. It is possible that subtle changes in gene expression were hard to detect when a small proportion of fibers are affected. It can also be that other pathways involved in myofiber atrophy contribute to the observed phenotype. Specific loss of type IIa and IIx fibers in muscles has been reported previously in aging and in FSHD [61,93,94,95,96,97]. Aging studies in 2-year-old murine muscle have shown that there is preferential loss of fiber size in type IIa and IIx fibers, with a 30% observed fiber loss, smaller type 2 fiber sizes, and a sparing of type IIb muscle fibers [93,94,95,96,97]. Expression profiling and proteomics of aging muscles showed increased metabolic- and mitochondrial-induced stress [93,94,97,98]. Recently, proteomic profiling of FSHD myoblasts showed that FSHD myoblasts contain a higher number of mitochondrial proteins in conjunction with reduced mitochondrial protein turnover [99]. This suggested that *DUX4* expression impacted the quality of the available mitochondrial proteins, even though the quantity was not reduced [99]. Impairments in quality control mechanisms of the mitochondria have been reported to lead to an excessive generation of reactive oxygen species, disorganization of the myofiber, and malformations in the mitochondria [88,97,100]. In humans, aging mechanisms and FSHD favor the preservation of the oxidative type I and loss of the glycolytic type II [61,93]. Muscle groups affected in FSHD, such as the zygomaticus major, orbicularis orbis, biceps brachii, triceps brachii, and tibialis anterior, contain a large proportion of fast twitch (type 2) fibers and a lower proportion of slow twitch (type 1) fibers [101,102,103,104,105]. Functional deficits and fiber size variation are common pathologies that are shared with the *ACTA-MCM/FLExDUX4* model, the *AAV-DUX4,* and *iDUX4pA* models [25,45,49,106,107]. However, the aggregation of TDP-43 in type IIb myofibers and preferential decrease in type IIa and IIx fiber sizes are new findings from characterizing the *FLExDUX4* mice [48,49,106,107]. To date, TDP-43 has only been shown to aggregate in the nucleus of FSHD myotubes in vitro but not in the cytoplasm [64,65]. Since humans were reported to not have type IIb fibers, this phenomenon may be unique to the mouse model but still worth further investigation [61,93,108,109].

Previous studies have reported the presence of inflammatory infiltrate, fibrosis, and fat deposition in affected patients with FSHD. The *FLExDUX4* mice presented with mild but significant increases in fat and fibrosis in response to chronic exposure to low levels of leaky *DUX4*. However, the low levels of *DUX4* expression do not induce severe pathologies, such as muscle necrosis and severe inflammation. When the *DUX4* is induced to be expressed at a higher level in the *ACTA1-cre/FLExDUX4* mouse model, the mice develop pathological changes, including degeneration/regeneration and inflammation similar to muscle pathologies seen in the affected areas in FSHD [48,49]. Please note that the less affected areas in muscles from individuals with FSHD also do not show overt pathology, which is similar to the observation in the *FLExDUX4* mice. Based on these findings, we propose that the *FLExDUX4* mice recapitulate the state when the muscles are mildly affected by the low level of DUX4 expression.

In the conditional *ACTA-MCM/FLExDUX4* models, continuous induction of the *DUX4* transgene is required to maintain what has been coined as moderate levels of pathology because the regenerated muscle fibers do not express equivalently high levels of DUX4 [25,49]. Continuous induction would be required to assess long-term therapeutic efficacy in a pre-clinical trial [106,107]. Of note, the uninduced *ACTA-MCM/FLExDUX4* mice do not have overt degeneration and regeneration [49]. However, the DUX4 protein was reported to be detected in uninduced *ACTA-MCM/FLExDUX4* mice, which was not detected in the *FLExDUX4* mice [48,49]. We show that the *FLExDUX4* model can reliably express *DUX4* mRNA up to 12 months old, which can be beneficial for testing agents directly targeting the *DUX4* transcripts [48,110,111]. This will be an advantage, as consistent DUX4 expression in the model reduces concerns from *DUX4* fluctuation due to the degeneration and regeneration of the muscles, such as in the *ACTA-MCM/FLExDUX4* model. Further study into the *FLExDUX4,* uninduced *ACTA1-MCM/FLExDUX4*, and induced *ACTA1-MCM/FLExDUX4* models to determine how different levels of *DUX4* affect the pathology of the muscles would help elucidate the disease mechanisms, as would the selection of proper models for specific types of therapeutic approaches.

## 5. Conclusions

Prolonged expression of a low level of *DUX4* led to reductions in muscle and body weights, as well as muscle strength, in the *FLExDUX4* mice. Males developed the disease phenotypes at a younger age in comparison to the female mice. Several molecular pathways, including pathways involved in oxidative stress response and fibrosis, were altered in the affected muscles. In addition, prolonged expression of DUX4 lead to preferential decreases in type IIa and IIx myofibers and TDP43-containing aggregates in type IIb myofibers. This study demonstrated that *FLExDUX4* mice have measurable phenotypes, some of which reflect human FSHD. The functional, pathological, and molecular changes can potentially be considered to be used as outcome measures when testing agents targeting DUX4.

## Figures and Tables

**Figure 1 jpm-13-01040-f001:**
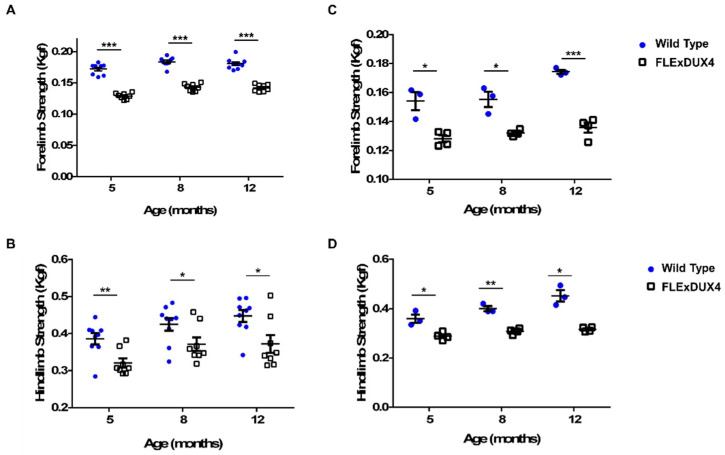
*FLExDUX4* mice display muscle weakness from 5 months to 12 months of age. Forelimb (**A**) and hindlimb (**C**) grip strength measurements of male wild-type (blue) and *FLExDUX4* (white) mice (Wild-type *n* = 9; *FLExDUX4 n* = 8). Forelimb (**B**) and hindlimb (**D**) grip strength measurements of female wild-type and *FLExDUX4* mice (Wild-type *n* = 3; *FLExDUX4 n* = 4). Error bars mark ± the standard error of the mean. * *p* < 0.05, ** *p* < 0.01, *** *p* < 0.001.

**Figure 2 jpm-13-01040-f002:**
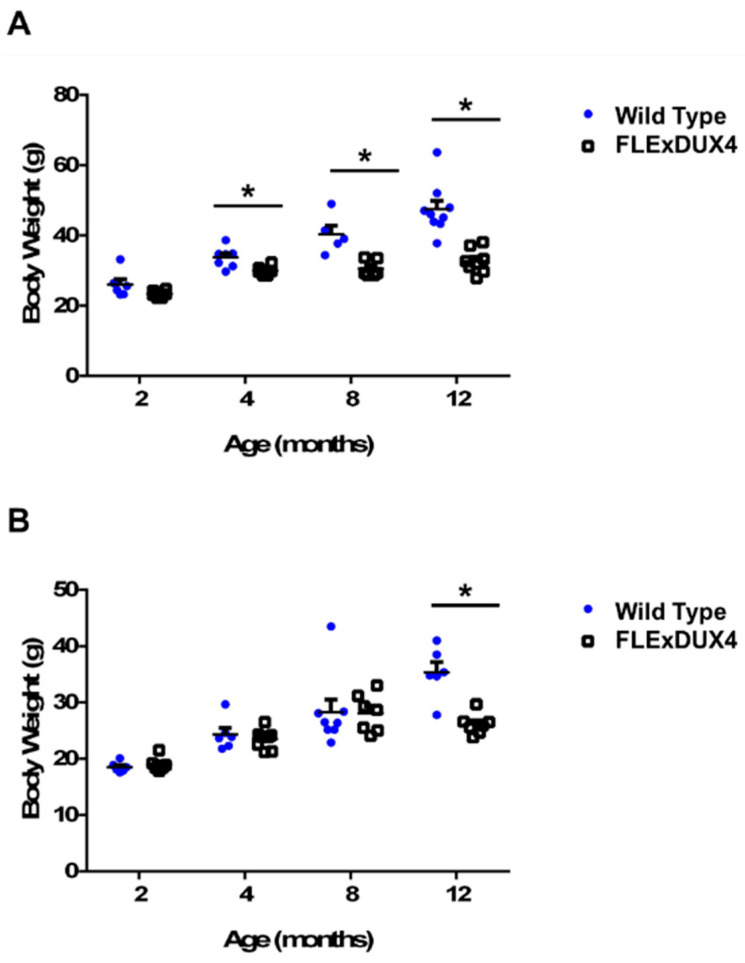
Male *FLExDUX4* mice showed body weight differences. (**A**) Male *FLExDUX4* mice showed body weight differences as early as 4 months of age. (**B**) Female *FLExDUX4* mice did not show body weight differences until 12 months of age. * *p* < 0.05. Error bars are ± the standard error of the mean.

**Figure 3 jpm-13-01040-f003:**
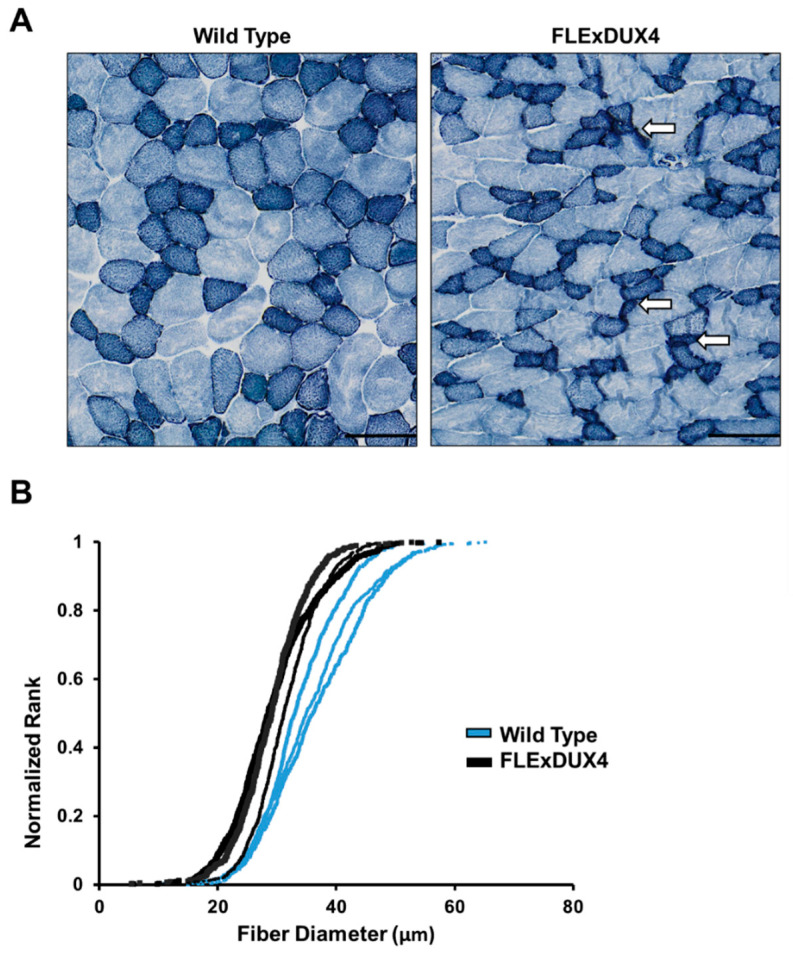
Small and angular-shaped NADH-TR-positive myofibers in quadriceps muscle of 12-month-old male *FLExDUX4* mice. (**A**) Cross-section of quadriceps stained with NADH-TR in wild-type and *FLExDUX4* mice. White arrows depict examples of small angular myofibers. (**B**) S-curve depicting the fiber sizes of NADH-TR-positive fibers of the *FLExDUX4* mice and their wild-type littermates, ranked by size. The scale bar marks a 100 µm distance. A total of 586–799 fibers were measured from the same area of the quadriceps muscles in each section (*n* = 3).

**Figure 4 jpm-13-01040-f004:**
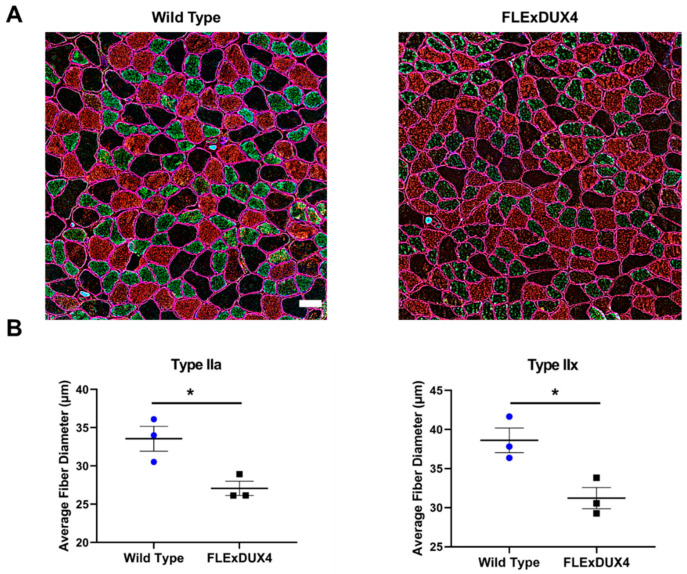
*FLExDUX4* mice had smaller type IIa and type IIx fibers at 12 months of age. (**A**) Type IIa (green) and type IIx (red) fibers were measured. Sarcolemma was stained by dystrophin staining (magenta). Scale bar measures 50 µm. (**B**) Type IIa (left) and type IIx (right) are significantly smaller in muscles of the *FLExDUX4* mice. The average fiber size of all measured fibers from each mouse (*n* = 3). Error bars mark the standard error. * *p* < 0.05. Error bars are ± the standard error of the mean.

**Figure 5 jpm-13-01040-f005:**
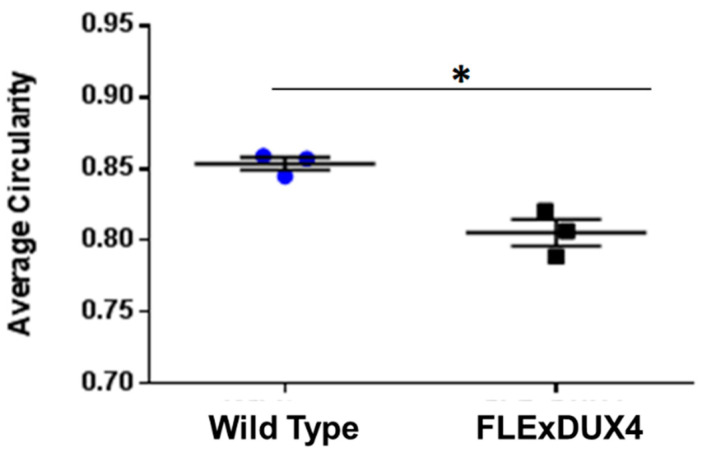
The circularity is reduced in type IIx fibers in *FLExDUX4* mice at 12 months of age. Error bars are ± the standard error of the mean. * *p* < 0.05.

**Figure 6 jpm-13-01040-f006:**
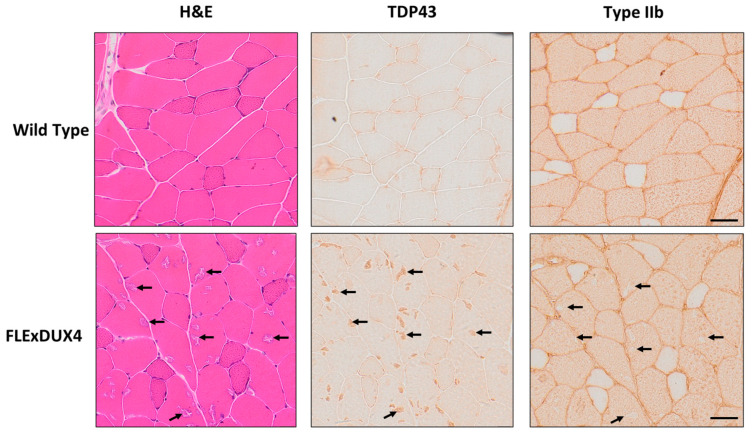
TDP-43-positive aggregates are in type IIb myofibers in quadriceps of 12-month-old male *FLExDUX4* mice. Serial sections of 12-month-old *FLExDUX4* male quadriceps were taken. Basophilic aggregates in sections stained with hematoxylin and eosin (H&E) were observed in the quadriceps of the *FLExDUX4* mice. The TDP-43-positive fibers are also stained positive for the myosin heavy chain type IIb. Black arrows denote examples of aggregates present in the same myofiber. Scale bar measures 50 µm.

## Data Availability

The RNA-seq data are deposited in NIH GEO database (GSE233832).

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
