# Peer review of "Molecular and Phenotypic Changes in FLExDUX4 Mice"

_jpm, 2023, doi:10.3390/jpm13071040_

Round 1

Reviewer 1 Report

The authors study FLExDUX4 mice that present leaky expression of the DUX4 transgene in the absence of induction. This low chronic DUX4 expression is expected to better reflect the pathological conditions of patients affected with FSHD. The authors study the evolution of mouse muscle strength, mouse body and muscle weight, muscle fiber types, size and histological structure.  Interestingly they show TDP43 aggregates forming over one year.

Comments:

The study is clean and well done on appropriate number of mice for statistical significance. The methods are very well described and the figures clearly display the data. The  manuscript correctly describes the experiments and conclusions, but is difficult to read because of multiple language mistakes.

I have the following questions.

1. A similar mouse model leading to low chronic DUX4 expression has been thoroughly investigated by Bosnakovski et al  2017 (ref 43). The present manuscript discussion should compare ref 43 results to those obtained in the present study and highlight any added value as well for muscle pathological evolution as for the altered signaling pathways detected in RNAseq data. 

2. Figure 3 does not convincingly demonstrate changes in fiber diameters of the FLExDUX4 mice versus controls

3. One of the major pathways disturbed in the present mouse model is adipogenesis as stated lines 428 and following. This suggests fat deposition in muscles. Because of its importance in patient muscle pathology the authors should investigate this point in their histochemical study of mouse muscle alteration over time. 

4. Line 547 and following : a more in depth discussion should be provided about the specific fiber type loss in the present study and its putative causes such as mitochondrial dysfunction

5. Line 36: it is not correct to state that FSHD 1 and 2 present different genetic mutations. Both present a DUX4 gene on the 4qA permissive allele with the polyadenylation signal allowing for toxic DUX4 protein expression. They only differ in the way the epigenetic condition i.e. hypomethylation on the D4Z4 repeat array, is obtained, either following a change in D4Z4 copy number (FSHD1) or resulting from loss of function in a gene encoding a protein involved in D4Z4 DNA methylation (FSHD2). FSHD1 has a monogenic transmission, while FSHD2 has a digenic transmission.

6. Line 119 and following: the whole method name is RT-qPCR (not qRT-PCR).  The primers used for the PCR step are provided in the Method section but there is no information about the qPCR quantification process and the equipment used: this should be added to the manuscript.  

7. The manuscript language should be corrected by a native English speaker. Similar mistakes occur multiple times about concord of subject and verb, use of same tense for several verbs in a given sentence (e.g. lines 463-464: showed ... leads), omission of "the" before proper nouns such as DUX4...

The mentions of the mouse ages are wrongly stated in most sentences (e.g. line 14: at 2 months old mice; line  351: "at 2-months-old") but intriguingly sometimes correct e.g. line 425: "2-month-old male mice . At 2 months of age" 

Minor comments

line 11: DUX4 gene expression is leaky (not leaked)

line 31:  use precise scientific term instead of "shin"

lines 33-35: avoid stating twice "individuals with FSHD" in the same sentence

line 57: additional post-natal tissues express DUX4: keratinocytes and mesenchymal stem cells

line 59: DUX4 expression occurs at the 4- to 8-cell stage in human embryos (it is mouse Dux that is expressed at the 2-cell stage)

line 63: a reference demonstrating DUX4 impact on myogenesis is missing: Knopp et al J. Cell Sci. 2016

line 74-76:  DUX4 gene expression is leaky, not the mice! But the mice do not develop overt muscle pathology, thus "which" should not refer to the DUX4 gene but to the mice, please clarify and correct the whole sentence.

line 115: 175-bp product, please correct everywhere in the text

line 156: use italics for Latin names of muscles

lines 163 and following: hind limb or hindlimb? fore limb or forelimb?

lines 293-294: avoid "age" twice in the same sentence

Line 319: avoid interpreting the results in figure legends ("showed body weight differences")

line 357: word missing after "were"

line 423: the paragraph number 3.4. is wrong

line 435: the official gene nomenclature uses different cases according to the species and should be in italics.  "Clock" is a mouse gene while "CLOCK" is a human one. Please check all gene names over the whole manuscript.

line 461: correct FELx to FLEx

line 480: briefly remind the reader why tamoxifen is used in this mouse model

lines 490-492: avoid repeating "for FSHD"in the same sentence

line 496: please clarify : "before the tamoxifen"

line 512: the reference is "Turki et al" not "Turkil"

lines 538-539: avoid repeating "aging muscles"in the same sentence

lines 570-572: please clarify the sentence

Author Response

Reviewer 1

  1. A similar mouse model leading to low chronic DUX4 expression has been thoroughly investigated by Bosnakovski et al 2017 (ref 43). The present manuscript discussion should compare ref 43 results to those obtained in the present study and highlight any added value as well for muscle pathological evolution as for the altered signaling pathways detected in RNAseq data.

Response: We added additional information of the model and the comparison in the discussion section, which is highlighted in yellow.

  1. Figure 3 does not convincingly demonstrate changes in fiber diameters of the FLExDUX4 mice versus controls

Response: We appreciate the comments and clarified the purpose of presenting the data using Figure 3 in the result section, which is now moved to supplemental figure section. In addition, we added the histogram to address the same point and to assist visualization.

  1. One of the major pathways disturbed in the present mouse model is adipogenesis as stated lines 428 and following. This suggests fat deposition in muscles. Because of its importance in patient muscle pathology the authors should investigate this point in their histochemical study of mouse muscle alteration over time.

Response: We included BODIPY lipid staining in the paper, which shows significantly higher fat deposition in muscles.

  1. Line 547 and following : a more in-depth discussion should be provided about the specific fiber type loss in the present study and its putative causes such as mitochondrial dysfunction

Response: We included more text discussing the relationship between specific fiber type loss and mitochondrial dysfunction.

  1. Line 36: it is not correct to state that FSHD 1 and 2 present different genetic mutations. Both present a DUX4 gene on the 4qA permissive allele with the polyadenylation signal allowing for toxic DUX4 protein expression. They only differ in the way the epigenetic condition i.e. hypomethylation on the D4Z4 repeat array, is obtained, either following a change in D4Z4 copy number (FSHD1) or resulting from loss of function in a gene encoding a protein involved in D4Z4 DNA methylation (FSHD2). FSHD1 has a monogenic transmission, while FSHD2 has a digenic transmission.

Response: We clarified that the genomic mutations affect the epigenetic control of the D4Z4 region, which lead to the de-repression of DUX4.  The modified section is highlighted in yellow.  We respectfully disagree that FSHD1 and 2 have the same genetic mutations.  While the downstream epigenetic defects and consequences are similar, the genomic mutations between FSHD1 and 2 are different, including different genomic regions and different types of mutations (e.g. contraction of a microsatellite repeat array and mutations in protein-coding genes).

  1. Line 119 and following: the whole method name is RT-qPCR (not qRT-PCR). The primers used for the PCR step are provided in the Method section but there is no information about the qPCR quantification process and the equipment used: this should be added to the manuscript.

Response: We edited the abbreviations in the paper and added the requested information.

  1. The manuscript language should be corrected by a native English speaker. Similar mistakes occur multiple times about concord of subject and verb, use of same tense for several verbs in a given sentence (e.g. lines 463-464: showed ... leads), omission of "the" before proper nouns such as DUX4...

The mentions of the mouse ages are wrongly stated in most sentences (e.g. line 14: at 2 months old mice; line 351: "at 2-months-old") but intriguingly sometimes correct e.g. line 425: "2-month-old male mice . At 2 months of age"

Response: This manuscript was written by a native English speaker and reviewed by a coauthor who is a native speaker in addition to the other co-authors. That said, we do agree with the reviewers’ critiques and appreciate all the valuable comments that significantly improve the readability of the manuscript. We carefully reviewed the manuscript to address the critiques, including subject/verb agreement, the mouse ages, and proper tenses in sentences.

line 11: DUX4 gene expression is leaky (not leaked)

Response: We corrected the word “leaked” to “leaky”

line 31: use precise scientific term instead of "shin"

Response: We changed “shin strengthening” to physical therapy.  While the term “shin strengthening” has been published in reviews, it is just one form of therapy recommended to patients with FSHD.  By using the term physical therapy and adding orthotics, the sentence now becomes more all-encompassing as each treatment plan is unique.

lines 33-35: avoid stating twice "individuals with FSHD" in the same sentence

Response: Edited the appropriate wording

line 57: additional post-natal tissues express DUX4: keratinocytes and mesenchymal stem cells

Response: The tissues were added to the text and cited in line 60

line 59: DUX4 expression occurs at the 4- to 8-cell stage in human embryos (it is mouse Dux that is expressed at the 2-cell stage)

Response: We corrected it to 4- to 8-cell stages.

line 63: a reference demonstrating DUX4 impact on myogenesis is missing: Knopp et al J. Cell Sci. 2016

Response: The appropriate citation for Knopp et al J. Cell Sci. 2016 was added in and is source number 43

line 74-76: DUX4 gene expression is leaky, not the mice! But the mice do not develop overt muscle pathology, thus "which" should not refer to the DUX4 gene but to the mice, please clarify and correct the whole sentence.

Response: We edited the appropriate section

line 115: 175-bp product, please correct everywhere in the text

Response: The corrections were made.

line 156: use italics for Latin names of muscles

Response: We appreciate this comment but italicizing the muscle names that are denoted in Latin is not a common practice in scientific journals, including the scientific articles we cited in this paper.

lines 163 and following: hind limb or hindlimb? fore limb or forelimb?

Response: We corrected all instances of “hind limb” and “fore limb” to “hindlimb” and “forelimb” respectively as the term is one word not two.

lines 293-294: avoid "age" twice in the same sentence

Response: We corrected the sentence in Figure 1.

Line 319: avoid interpreting the results in figure legends ("showed body weight differences")

Response: We corrected the legend.

line 357: word missing after "were"

Response: The word “determined” was added after “were”.

line 423: the paragraph number 3.4. is wrong

Response: The error was corrected with the appropriate heading number 3.5.  Additionally, the heading format was also corrected in the manuscript

line 435: the official gene nomenclature uses different cases according to the species and should be in italics. "Clock" is a mouse gene while "CLOCK" is a human one. Please check all gene names over the whole manuscript.

Response: We double checked our use of italics.  Everything not italicized was to intentionally refer to the protein.  For example, using “Clock” because the protein Clock activates the NRF2 pathway.  We also corrected the capitalized CLOCK, it was used as written in the source cited despite the work being done in mice (“CLOCK and BMAL1 regulate MyoD and are necessary for maintenance of skeletal muscle phenotype and function”).

line 461: correct FELx to FLEx

Response: The typo was corrected

line 480: briefly remind the reader why tamoxifen is used in this mouse model

Response: We chose to remind the reader about tamoxifen in line 496 as this paragraph describes the results from the FLExDUX4 mice and not the ACTA-MCM/FLExDUX4 line which can be induced with tamoxifen.

lines 490-492: avoid repeating "for FSHD"in the same sentence

Response: The duplicated words were removed

line 496: please clarify : "before the tamoxifen"

Response: The sentence was reworded to remind the readers that tamoxifen induces a more robust DUX4 expression compared to the FLExDUX4 and ACTA-cre/FLExDUX4 mice

line 512: the reference is "Turki et al" not "Turkil"

Response: The in-text reference of the author’s name has been corrected

lines 538-539: avoid repeating "aging muscles"in the same sentence

Response: The duplicate wording was removed

lines 570-572: please clarify the sentence

Response: We reread the sentence and clarified the section.

Reviewer 2 Report

Regarding ms jpm-2320277 Murphy et al.

This manuscript is a study of the FLExDUX4 mouse and the muscle phenotype and pathology of FSHD in aged mice.

Concerns that should be addressed are below.

1. The FLExDUX4 mice as engineered by Murphy et al., allow for Cre mediated rearrangement of the transgene to express DUX4. The tamoxifen inducible Acta1-MCM Cre transgene is leaky, in the absence of Tamoxifen, there is a certain amount of Cre recombinase that is expressed. This is known for many tamoxifen inducible Cre transgenes. What is unclear to me is how the FLExDUX4 transgene, without a Cre present in the genome, is ‘leaky’. I think that the authors need to answer the following:

1.1 What causes the rearrangement of the transgene in the Rosa locus to allow for expression of DUX4?

1.2 What evidence do the authors have to prove that there is rearrangement of the transgene as well as the subsequent expression of DUX4?

1.3 How consistent is this rearrangement across different muscles? What level of rearrangement is required for FSHD phenotypical changes to occur?

2. Figures 3 and 4 should be presented as distributions, the data as presented is not interpretable. Further for Fig. 3, not only should the number of animals be reported but the number of fibers examined for each animal and in total by genotype.

3. Fig. 4 the number of NADH-TR myofibers should be reported or each fiber size.

4. It is unclear what an analysis of circularity is providing, especially in light of fixed muscle tissue that will have artifacts.

5. RNA-Seq data should be available in a public repository and the accession information provided.

6. Grip strength does not correlate with muscle weight, but atrophy is the stated mechanism of FSHD defects, the authors should address this.

Author Response

Reviewer 2

  1. The FLExDUX4 mice as engineered by Murphy et al., allow for Cre mediated rearrangement of the transgene to express DUX4. The tamoxifen inducible Acta1-MCM Cre transgene is leaky, in the absence of Tamoxifen, there is a certain amount of Cre recombinase that is expressed. This is known for many tamoxifen inducible Cre transgenes. What is unclear to me is how the FLExDUX4 transgene, without a Cre present in the genome, is ‘leaky’. I think that the authors need to answer the following:

1.1 What causes the rearrangement of the transgene in the Rosa locus to allow for expression of DUX4?

1.2 What evidence do the authors have to prove that there is rearrangement of the transgene as well as the subsequent expression of DUX4?

1.3 How consistent is this rearrangement across different muscles? What level of rearrangement is required for FSHD phenotypical changes to occur?

Response: We much appreciate the reviewer’s comments and we clarified the texts in both introduction and discussion sections. The FLExDUX4 model was previously generated by Drs. Takako and Peter Jones at the University of Nevada Reno School of Medicine and the University of Massachusetts Medical School.  Because of the association with the RosA26 promoter, it was a possibility that the DUX4 transgene would be ubiquitously expressed and had the possibility to be toxic to the animals.  To prevent this, the transgene was inverted.  Cross breeding with the ACTA1-MCM mice would allow cre-recombinase to be expressed in a skeletal muscle specific manor, but would only activate in the presence of tamoxifen due to being flanked by mutated estrogen receptors.  However; in both of the papers published on the model (“A cre-inducible DUX4 transgenic mouse model for investigating facioscapulohumeral muscular dystrophy” and “Transgenic mice expressing tunable levels of DUX4 develop characteristic facioscapulohumeral muscular dystrophy-like pathophysiology ranging in severity”) and in our paper, it was shown that the DUX4 mRNA is transcribed (leaky) in the absence of recombination.

The reason for this leakiness is currently unknown and other models have displayed leakiness using the same gene.  Aberrant recombination was already discussed by Jones and Jones in “A cre-inducible…” and they found that it was not the cause of the leaky transgene.  Additionally, it has been shown that the FLExDUX4, uninduced ACTA1-MCM/FLExDUX4, and induced ACTA1-MCM/FLExDUX4 mice all express similar levels of DUX4 mRNA, but have varying degrees of detectable DUX4 protein (“Transgenic mice expressing tunable levels of DUX4 develop characteristic facioscapulohumeral muscular dystrophy-like pathophysiology ranging in severity”).

In our paper, we focused purely on the FLExDUX4 model which does not carry the ACTA1 controlled cre-recombinase.  We clarified this point in the manuscript further to reduce confusion.

  1. Figures 3 and 4 should be presented as distributions, the data as presented is not interpretable. Further for Fig. 3, not only should the number of animals be reported but the number of fibers examined for each animal and in total by genotype.

Response: We added supplemental figures with histograms to address the reviewers’ comments and to complement the current figures. The animal numbers and numbers of fibers measured across of each section were reported in the Materials and Methods section as well as in the figure legends. We also clarified the text of the results section to discuss the changes that were observed.

We chose to present the data in an S-curve sorted by rank because it provides a clear view of the distributions of each individual mouse.  The purpose of the graph was to illustrate that while there were no obvious overall changes that were observed, the key differences lay in the smaller fibers. To avoid confusing readers, we moved it to supplemental figure section.

  1. Fig. 4 the number of NADH-TR myofibers should be reported or each fiber size.

Response: The number of NADH-TR positive fibers measured were reported in the Materials and Methods section and in the figure legend.

  1. It is unclear what an analysis of circularity is providing, especially in light of fixed muscle tissue that will have artifacts.

Response: Circularity is a measure of how round a shape is where a value of 1 denotes a perfect circle.  This measurement is very helpful in quantifying angular fibers.  Healthy myofibers should maintain a high value in circularity as they appear fairly round.  Atrophic fibers will have lower circularity values since they are frequently more angular and triangular shaped.  The concern about artifacts was a concern while we were processing our samples, since improper sectioning would also impact the fiber diameters.  We did visually evaluate the integrity of the section before quantifying both the myofiber diameter and the circularity of the sections.  For example, if the muscle was not perpendicular to the cryostat blade, the cross section would be uniformly full of long angular fibers that would artificially impact the fiber size and circularity of the fibers.  We clarified the rationale of performing this analyses is the manuscript, highlighted in yellow.

  1. RNA-Seq data should be available in a public repository and the accession information provided.

Response: The data has been deposited and the GEO accession number is included.

  1. Grip strength does not correlate with muscle weight, but atrophy is the stated mechanism of FSHD defects, the authors should address this.

Response: Multiple studies have shown that a variety of conditions can impact grip strength in mice.  We addressed the reviewer’s comment in the discussion section, which is highlighted in yellow. One condition to affect muscle function is the quality of the muscle.  Molecular changes in metabolism, fibrosis, and atrophy have been previously reported to affect grip strength.  In this paper, we reported that specifically type 2a and 2x fibers were affected.  These fibers are more oxidative compared to the type 2b fibers.  Changes in the metabolism of these fibers could contribute to muscle weakness.  We cited three papers in this study that show that subtle changes are sufficient to impact grip strength measurements.  We have clarified the point in the manuscript.

Round 2

Reviewer 1 Report

The authors have  appropriately dealt with most of my comments as well in their answers as in the revised version of their manuscript.

However, I still have one scientific comment and a few typo/minor spell check requests

Scientific comment: I previously requested an additional paragraph in the discussion on the data obtained by another group on the iDUX4pA mouse model. The authors have correctly discussed the 2017 publication of that group in Nature Communication (ref 45 in the revised manuscript) about stochastic low level DUX4 expression in these mice. However the authors have omitted to refer to the next publication from the same group (Bosnakovski et al 2020 J. Clin. Invest.) on the same mouse model followed for a longer period of time (6 months) and that presented high similarities with FSHD muscle pathology.  This 2020 publication should be added to the reference list in the manuscript, and because it investigated long term /low DUX4 expression similar to the present study on FLExDUX4 mice, the results obtained on both mouse models should be compared in the present manuscript discussion. 

Typo/minor spell check requests:

Line 20: suppress "a" (the verb next line is plural)

Line 387: fiber type IIa, not Iia

Line 422: fiber type IIb, not Iib

Line 430: avoid repeating "type IIb" in the same sentence

Line 495: suppress "the" in front of "DUX4"

Line 506: same mistake as stated in my first review about age mention! All these errors were corrected in the revised manuscript, please do it as well in the added yellow paragraphs (by the way, the supplementary figure legends were fine)!

Correct to "reported to live until 4 months of age"

Line 889: please mention the mouse ages in Suppl Fig3 as done in Suppl Fig2

Author Response

Scientific comment: I previously requested an additional paragraph in the discussion on the data obtained by another group on the iDUX4pA mouse model. The authors have correctly discussed the 2017 publication of that group in Nature Communication (ref 45 in the revised manuscript) about stochastic low level DUX4 expression in these mice. However the authors have omitted to refer to the next publication from the same group (Bosnakovski et al 2020 J. Clin. Invest.) on the same mouse model followed for a longer period of time (6 months) and that presented high similarities with FSHD muscle pathology.  This 2020 publication should be added to the reference list in the manuscript, and because it investigated long term /low DUX4 expression similar to the present study on FLExDUX4 mice, the results obtained on both mouse models should be compared in the present manuscript discussion. 

Response:

          We added the citation and included more sentences to compare the models. The added sentences are highlighted in yellow.

Typo/minor spell check requests:

Line 20: suppress "a" (the verb next line is plural)

Response: Corrected the verb agreement

Line 387: fiber type IIa, not Iia

Response: Corrected the typo

Line 422: fiber type IIb, not Iib

Response: Corrected the typo

Line 430: avoid repeating "type IIb" in the same sentence

Response:

We removed the type IIb

Line 495: suppress "the" in front of "DUX4"

Response:

We removed “the” from the sentence.

Line 506: same mistake as stated in my first review about age mention! All these errors were corrected in the revised manuscript, please do it as well in the added yellow paragraphs (by the way, the supplementary figure legends were fine)!

Correct to "reported to live until 4 months of age"

Response:

          We made the corrections.

Line 889: please mention the mouse ages in Suppl Fig3 as done in Suppl Fig2

Response:

          We added the mouse ages in Suppl Fig3.

Reviewer 2 Report

I much improved manuscript. I have only one point that I disagree with the comments by the authors on. Regarding the circularity data, artifacts of fixation are rarely uniform or repeatable, so these data have dubious value.

Author Response

I much improved manuscript. I have only one point that I disagree with the comments by the authors on. Regarding the circularity data, artifacts of fixation are rarely uniform or repeatable, so these data have dubious value.

Response: Reviewer’s point is noted.  We conducted frozen sections to avoid issues associated with fixation. We hope this will resolve the specific concern.